# An HLD Model for Tomato Bacterial Canker Focusing on Epidemics of the Pathogen Due to Cutting by Infected Scissors

**DOI:** 10.3390/plants11172253

**Published:** 2022-08-30

**Authors:** Akira Kawaguchi, Shoya Kitabayashi, Koji Inoue, Koji Tanina

**Affiliations:** 1Western Region Agricultural Research Center (WARC) (Kinki, Chugoku, and Shikoku Regions), National Agriculture and Food Research Organization (NARO), Fukuyama 721-8514, Japan; 2Research Institute for Agriculture, Okayama Prefectural Technology Center for Agriculture, Forestry and Fisheries, Akaiwa City 709-0801, Japan; 3Okayama Agriculture Development Institute, Akaiwa City 701-2221, Japan

**Keywords:** *Clavibacter michiganensis* subsp. *michiganensis*, tomato bacterial canker, HLD model, SIR model, disinfection of pruning shears

## Abstract

A healthy, latently infected, diseased (HLD) plant model for botanical epidemics was defined for tomato bacterial canker (TBC) caused by the pathogenic plant bacteria, *Clavibacter michiganensis* subsp. *michiganensis* (*Cmm*). To estimate the infection probability parameter, inoculation experiments were conducted in which it was assumed that infection is transferred to healthy plants through contaminated scissors used to cut symptomless infected plants. The approximate concentration of *Cmm* in symptomless infected plants was 1 × 10^6^ cells/mL, and the probability of infection of healthy tomato plants was approximately 0.75 due to cutting with scissors soaked in a cell suspension of *Cmm* at 1 × 10^6^ cells/mL. Three different HLD models were developed by changing some parameters, and the D curve calculated by the developed HLD model A was quite similar to the curve of the proportion of diseased plants observed in fields that had a severe disease incidence. Under a simulation of disease incidence using this model, the basic reproduction number (*R*_0_) was 2.6. However, if the infected scissors were disinfected using ethanol, *R*_0_ was estimated as 0.3. The HLD model for TBC can be used to simulate the increasing number of diseased plants and the term of disease incidence.

## 1. Introduction

Tomato bacterial canker (TBC), caused by *Clavibacter michiganensis* subsp. *michiganensis* (*Cmm*), is one of the most important tomato diseases and causes substantial economic loss worldwide, including Japan [1,2,3,4,5]. Sources of primary *Cmm* inoculum are infected seeds, infected transplants, and infected tomato debris [2,6,7,8]. Secondary spread is caused by splashes of water, non-sterilized pruning shears, or workers’ hands [2,8,9,10,11]. TBC causes frequent outbreaks in commercial greenhouses, where cutting lateral buds of tomato plants contributes to its secondary spread [2,10]. *Cmm* is exuded through hydathodes in guttation fluid under high-humidity conditions, such as inside greenhouses [11]. In commercial greenhouses, TBC outbreaks in which the number of diseased plants increases rapidly after cutting are often observed. Thus, the authors previously verified that the spread of *Cmm* by cutting occurred sooner than that by infection from tomato debris [8]. The source of bacterial canker infection in greenhouses in Japan consisted of residual plants in the soil as the primary inoculum and disbudding and defoliation with contaminated equipment and hands as the secondary inoculum [2]. Infection spreads rapidly due to transfer of *Cmm* to nearby healthy plants by disbudding and defoliation [2,8]. Therefore, farmers are unwittingly cutting latently infected tomato plants [8]. To prevent outbreak of TBC in commercial fields, disinfection of agricultural equipment, such as pruning shears and gloves, before cutting is important. 

Although these previous reports help us understand the epidemics of TBC, the disease is still causing frequent outbreaks around Japan [8]. Therefore, we need to precisely analyze epidemics of TBC using another approach. The mathematical theory and modelling of epidemics, such as the SIR model [12], is common in the research on infectious disease epidemics [13]. Though the SIR model is a classic and simple theory [12], it was recently used for epidemiological studies of various infectious diseases in human and animals, such as COVID-19 [13,14]. In botanical epidemiology, SIR and other related models have been used to understand the mechanisms of plant disease epidemics [15,16,17,18]. However, there has been no epidemic model for TBC using a SIR model theory until now. Thus, the objectives of this study were to develop the HLD epidemic model based on the SIR theory of TBC and to analyze the relationship between disease incidence and infection probability by cutting plants with infected scissors. In addition, we assessed the efficacy of disinfection of scissors contaminated with *Cmm* to prevent infection of *Cmm* by cutting.

## 2. Results

### 2.1. Duration of Infection and Effect of Ethanol Disinfection of Infected Scissors

In Table 1, the intervals between inoculation, by cutting non-disinfected plants with infected scissors, and symptom development were 21, 11, 7, and 12 days in each experiment. The mean was 12.0 (95% confidence interval (CI): 7.5–16.5). Thus, that value was defined as a parameter of the duration of infection (*d*) (Table 2). Based on Kaplan-Meier survival analysis, in four independent experiments, ethanol disinfection of infected scissors resulted in fewer diseased plants in a longer period with significantly more survival days than non-disinfection of infected scissors (Figure 1A, Table 1). On average, ethanol disinfection of infected scissors prolonged mean survival by 37.5 days compared to non-disinfection (Table 1). Meta-analysis of the four independent greenhouse experiments described above showed that the IRR was 0.12 (95% CI: 0.06–0.22, *p* ≤ 0.0001) (Figure 1B), indicating that the disease incidence was significantly reduced by ethanol disinfection. The heterogeneity of studies was tested in each meta-analysis. The *I^2^* value and *p* value were 35% and 0.20, respectively (Figure 1B), indicating the absence of heterogeneity among experiments.

### 2.2. Populations of Cmm in Tomato Plants with Different TBC Symptoms and Probability of Infection

In the symptomless but infected tomato plants, 6.47 log_10_ CFU/g plant tissue (95% CI: 5.75–7.20), which means approximately 3.0 × 10^6^ CFU/g plant tissue was detected (Figure 2A). In plants with early wilt symptoms, 7.38 log_10_ CFU/g (95% CI: 6.98–7.77), which means approximately 2.4 × 10^7^ CFU/g plant tissue, was detected (Figure 2A), and 6.61 (95% CI: 6.36–6.85) log_10_ CFU/g, which means approximately 4.4 × 10^6^ CFU/g, was detected in plants with late wilt symptoms (Figure 2A). The values of log_10_ CFU/g plant tissue with three different symptoms were not significantly different (*p* > 0.05), indicating that populations of *Cmm* in infected plants would range between 10^6^ and 10^7^ CFU/g plant tissue (Figure 2A).

Tomato plants were inoculated with four different concentrations of a *Cmm* cell suspension (10^5^, 10^6^, 10^7^, and 10^8^ cells/mL). The proportion of diseased plants at 10^5^ cells/mL of *Cmm* was significantly lower than that of other concentrations of cell suspension (Figure 2B). Proportions of diseased plants (%) were 70.3 (95% CI: 54.1–86.6), 75.0 (95% CI: 67.5–82.5), and 80.0 (95% CI: 61.3–98.7) at 10^6^, 10^7^, and 10^8^ cells/mL of *Cmm*, respectively, and there was no significant difference among the three plots (Figure 2B). Considering the results shown in Figure 2B, the probability of infection, which is a parameter of the infection rate per contact (*B*), was defined as 75/100, which was obtained from inoculation with a cell suspension of 10^7^ cells/mL of *Cmm* (Table 2).

### 2.3. Development of HLD Model

The parameters to develop the HLD model of TBC were estimated from data obtained in this study (Table 1, Figure 2A,B) and defined in Table 2. Based on the variables (*B* = 75/100, *D* = 12, *k* = 2/7, *γ* = 1/12) in Table 2, the dynamics of healthy plants (*H)*, latently infected plants (*L)*, diseased plants (*D)*, and *R_t_* are shown in HLD model A (Figure 3A). Moreover, two types of HLD models, B and C, based on the minimum (*B* = 67.5/100, *D* = 7.5, *k* = 2/7, *γ* = 1/7.5) and maximum (*B* = 82.5/100, *D* = 16.5, *k* = 2/7, *γ* = 1/16.5) variables within an assumed range using 95% CI are shown in Figure 3B,C, respectively. In model A, *R*_0_ was estimated as 2.6, and it could take 61 days from the beginning of transmission to decrease by *R_t_* < 1.0 (Figure 3A). In model B, *R*_0_ was estimated as 1.4, and it could take 100 days from the beginning of transmission to decrease by *R_t_* < 1.0 (Figure 3B). In model C, *R*_0_ was estimated as 3.9, and it could take 50 days from the beginning of transmission to decrease by *R_t_* < 1.0 (Figure 3C). Compared with three types of HLD models A, B, and C (Figure 3A–C) and the observed disease incidence in commercial greenhouses (Figure 3D), the curved shape of the dynamics of *D* in model A was close to that of observed disease incidence (Figure 3A–D). At 108 days after infection, model A showed a disease incidence of 876 of *R* per 1000 of *N*, and the observed data in a commercial greenhouse showed an average disease incidence of 84.1% (Figure 3A–D).

When infected scissors were disinfected with ethanol, parameter *B* was multiplied by the IRR value (=0.12) obtained from a meta-analysis (Figure 1B). Then, *R* was minimally increased in model A (Figure 3E), and *R*_0_ was calculated as 0.3, indicating that ethanol disinfection strongly inhibited the transmission of *Cmm* by scissors in greenhouses.

## 3. Discussion

Our previous report revealed that cutting using scissors infected with *Cmm* likely contributed considerably to frequent outbreaks of TBC [8]. In this study, we developed a disease-increasing simulation model of TBC based on an SIR model using several parameters that were newly defined by several experiments, and the model was able to estimate the dynamics of disease increase due to the use of infected scissors. In the initial condition of this model, the first latently infected plant (*L* = 1) was assumed to have occurred due to a primary inoculum, which was soil-borne or seed-borne. If a large amount of infected plant debris in the greenhouse or unwitting planting of infected nursery tomato plants is assumed, *L* should be >1. 

When statistical models are developed, big data like meteorological and/or disease outbreak data, which are commonly collected by governmental offices, will be used to estimate parameters [19,20,21,22,23,24,25]. However, there was no scientific evidence or reports about infection of *Cmm* occurring in commercial greenhouses in Japan. Thus, parameters were assumed in several actual experiments, in this study. An SIR model is used in human and animal diseases [13,14] and disease transmission from an infected person to a susceptible one is assumed to be by contact. In the case of plant diseases, assuming the parameter of contact times per day (*k*) is difficult. In this study, the *k* of TBC occurring in greenhouses was focused on growers’ contact as vectors of *Cmm* [2,8,10], and *k* could be assumed, but the *k* could be changed based on numbers of plants and/or several sizes of commercial greenhouses. The HLD models A, B, and C are not only applicable under several levels of TBC contamination in commercial greenhouses, if necessary, to change some parameter values, but effects of some control methods could also be simulated in HLD models.

The HLD model A closely matched the dynamics of a severe outbreak of TBC in actual commercial greenhouses. Model B showed a much lower curve of *R* and *R*_0_ values (=1.4) than those of model A, indicating that model B could be used for a mild outbreak of TBC, such as under low infection pressure. Model C showed a higher curve of *H* and *R*_0_ value (=3.9) than that of model A, indicating that model C could be used for much more severe outbreaks of TBC than expected. Although model A closely matched the dynamics of observed disease increases, these three models could be used for different levels of TBC outbreak in commercial greenhouses. Focusing on model A, *R*_0_ was estimated as 2.6, indicating that TBC is a high-risk plant disease due to the high value of *R*_0_, which was much greater than 1.0. The *R*_0_ value 2.6 showed that *Cmm* could be transmitted from one infected plant to 2.6 healthy plants in the initial condition. The high value of *R*_0_ in this study strongly supports the previous report that pruning using scissors infected with *Cmm* likely contributes considerably to frequent outbreaks of TBC [8]. In a human disease, the most important uses of *R*_0_ are determining if an emerging infectious disease can spread in a population and determining what proportion of the population should be immunized through vaccination to eradicate it [20,21]. In the case of plant diseases, immunization could be difficult, but specific tomato cultivars that are resistant to *Cmm* could be used [1]. Although there are a few cultivars of rootstock with mild resistance to *Cmm*, we have no scions that are strongly resistant. Thus, management using bactericides and/or disinfection methods is needed to bring the value of *R*_0_ down to 1.0.

In this study, approximately 3.0 × 10^6^ CFU/g plant tissue of *Cmm* was detected in the symptomless infected tomato plants, and TBC occurred in tomato plants cut by scissors infected with *Cmm* of 10^6^ and 10^7^ cells/mL, indicating that scissors infected by cutting symptomless infected plants could allow *Cmm* transmission to healthy plants and the onset of infection. Moreover, cutting with scissors infected with *Cmm* of 10^5^ cells/mL (which is much lower than 10^6^ cells/mL) infected plants, indicating that even low-level contamination of equipment and symptomless infected plants could carry a positive risk of *Cmm* transmission. This study revealed the relationship not only between the population of *Cmm* and probability of disease onset but also the risk of *Cmm* transmission by scissors contaminated by symptomless infected plants. 

Regardless of whether plants have symptoms or not, disinfection of scissors after cutting is needed to prevent TBC outbreak. In this study, the positive effect of ethanol disinfection (IRR value = 0.12) was also revealed. Theoretically, based on the result of the simulation in model A, ethanol disinfection could almost perfectly prevent disease spread and incidence, and the *R*_0_ was 0.3. Based on these results, it is recommended that growers sterilize their scissors and gloves by dipping them in 70% ethanol to prevent outbreaks of TBC in commercial greenhouses. Practically speaking though, this method might not be perfect. Growers might not perfectly disinfect scissors before every contact during a cultivation period. However, even if not perfect, disinfection is still recommended for TBC control based on the results of the four experiments in this study. Cutting with disinfected scissors could clearly extend the cultivation periods of healthy plants more than cutting with non-disinfected ones. In general, using non-*Cmm*-contaminated tomato seeds and maintaining nurseries by exhaustive inspections using several serological and/or molecular techniques are needed to prevent TBC incidence [1,8]. However, a small number of contaminated seeds and/or plants may slip through inspections based on false-negative results. Thus, ethanol disinfection should be conducted to prevent disease spread and incidence because even a very few contaminated seeds can be inocula in fields.

## 4. Materials and Methods

### 4.1. Duration of Infection and Effect of Ethanol Disinfection of Infected Scissors

The *Cmm* strain CMM16-3, which was isolated from tomato plants with TBC symptoms in a commercial field in Hiroshima, Japan, and was pathogenic against tomato (*Solanum lycopersicum* cv. Momotaro) was used in this study. Tomato plants (1 1/2 months old) grown from seeds were inoculated by snipping off compound leaves with infected scissors. Cell suspensions (adjusted OD_600_ = 0.1; 1 × 10^8^ cells/mL) of the *Cmm* strain were prepared from cultures grown for 48 h on potato dextrose agar medium. For the first inoculation method, scissors were dipped into the cell suspension for 1 s, and tomato plants growing in plastic pots with soil (20 cm depth × 9 cm diameter; 1 plant per pot) were inoculated by pruning two compound leaves. For the second inoculation method for assessment of disinfection, scissors were dipped into the *Cmm* suspension described above for 1 s, and the scissors were dipped into 70% ethanol solution for 1 s. After that, two compound leaves of each tomato plant were cut with the disinfected scissors. Inoculated plants were grown in a greenhouse at 25–28 °C, and disease symptoms (progressive wilting followed by leaf necrosis or tip dieback) were recorded every 2 or 3 days for around 2 months. The experiment was repeated four times (experiments 1 to 4) as independent replicates. Eighteen tomato plants were inoculated by each method in experiment 1, and 45 plants were inoculated by each method in experiments 2, 3, and 4, respectively. 

This analysis was performed using the RStudio user interface (RStudio, Inc., version 1.2.5001, Boston, MA, USA) for R software (R Foundation for Statistical Computing, version 3.6.1, Vienna, Austria). The R package “survival” was used to carry out the Kaplan-Meier survival analysis and the log-rank test [8,26]. Meta-analysis by the DerSimonian-Laird method was performed using the R package “metafar”. The effect size of treatment was calculated as an integrated relative risk (IRR), which was defined as estimate of the effect calculated using relative risk values and weightings in each experiment by the DerSimonian-Laird method [27,28,29].

### 4.2. Populations of Cmm with Different TBC Symptoms in Tomato Plants 

Tomato plants (1 1/2 months old) were grown from seeds for inoculation with *Cmm*. One cell suspension of *Cmm* (10^6^ cells/mL) was prepared. Scissors were dipped into the cell suspensions for 1 s, and 50 tomato plants were inoculated by cutting two compound leaves. Inoculated plants were grown in a greenhouse at 25–30 °C. To confirm that plants had been infected with *Cmm* at 7 days after inoculation (dai), ImmunoStrip for *Cmm* (Agdia, Elkhart, IN, USA), a rapid immune-chromatographic strip test, was used [30]. As with plants in symptomless plots, 10 plants without symptoms of TBC but with positive reactions to ImmunoStrip for *Cmm* were randomly chosen from inoculated plants. At 14 dai, as with plants in the early-wilt symptom plot, the other 10 plants with disease symptoms (progressive wilting followed by leaf necrosis or tip dieback) were randomly chosen from the remaining inoculated plants. At 3 months after inoculation (mai), as with plants in the late wilt symptom plot, 10 plants with tip dieback and naturally dried were randomly chosen from remaining inoculated plants as plants in the late wilt symptom plot. 

The populations of *Cmm* in selected 30 plants (10 plants at 7 dai, 10 at 14 dai, and 10 at 3 mai) were assessed by the serial dilution plate method [8]. Stem samples were collected that included a growing point of plants (0.2 g fresh weight per plant, 1 sample per plant). The samples were washed with sterile distilled water and then crushed in 1 mL of sterile distilled water using an autoclaved mortar and pestle. Ten-fold serial dilutions (100 μL) of the samples were then plated and spread on the *Cmm*-selective medium SMCMM [2,31] and the plates were incubated at 25 °C for 5 d. Colony growth was then observed on five plates for each dilution and the numbers of colony-forming units (CFU) were counted. The bacterial populations in plants (CFU/g of plant tissue) were log_10_-transformed before statistical analysis. This assay was independently performed twice. An ANOVA test (*n* = 20, 10 plants per experiment) was performed using RStudio.

### 4.3. Probability of Infection with TBC

Tomato plants (1 1/2 months old) grown from seeds were inoculated by snipping off compound leaves with infected scissors. Four different concentrations of cell suspensions of *Cmm* (10^5^, 10^6^, 10^7^, and 10^8^ cells/mL) were prepared. Scissors were dipped into one of the four different *Cmm* cell suspensions for 1 s, and tomato plants were inoculated by cutting two compound leaves. Inoculated plants were grown in a greenhouse at 25–30 °C, and disease symptoms (progressive wilting followed by leaf necrosis or tip dieback) were recorded 25 days after inoculation. The experiment was repeated 4 times (experiments 1 to 4). Sixty-four tomato plants for each different inoculation concentration were used in each experiment. ANOVA and Tukey’s multiple test (*n* = 4) were performed using RStudio.

### 4.4. Development of HLD Model

In this study, the HLD model was defined based on the SIR model. The TBC system can be modelled by a set of linked differential equations for a compartmental system to describe the change in status of tomato plants from healthy (*H*) to latently infected (*L*) (symptomless of TBC) and ultimately to diseased (*D*) (with symptoms of TBC) including canker, wilt, and dead plants. In commercial greenhouses, farmers are unwittingly cutting latently infected tomato plants, and latently infected plants are infectious [8]. In commercial greenhouse, *Cmm* transmission is mainly caused by non-sterilized pruning shears and/or workers’ hands [2,10]. Although the diseased plants with symptoms could be infectious, farmers avoid touching diseased plants with obvious symptoms and remove them from the greenhouse [2,8,10]. Thus, in this study, *D* was defined as post-infectious individuals. Comparing the definitions of HLD and SIR, *H*, *L*, and *D* correspond to *S*, *I*, and *R*, respectively. 

The general forms of the equations are given by:*dH*/*dt* = −*βHL*,(1)
*dL*/*dt* = *βHL* − *L*/*d*,(2)
*dD*/*dt* = *L*/*d*,(3)
*β* = *Bk*/*N*,(4)
*N* = *H* + *L* + *D*,(5)
in which *N* is the total number of tomato plants, *β* is the infection rate of plants, *B* is the infection rate per contact, *k* is the number of contacts per day, and *d* is the duration of infection (days). In this study, some parameters were defined by the common conditions of growing tomato plants in commercial greenhouses in Japan. The number of tomato plants in a commercial greenhouse is commonly about one thousand. Thus, in the beginning of infection in a greenhouse, *N* was 1000 tomato plants (*H* = 999, *L* = 1, *D* = 0). The first infected plant, *L*, was assumed to have been due to soil infection or seed borne infection. In general, the growers conduct disbudding approximately twice and defoliation once on one plant in a week. Thus, *k* was defined as 2/7 (≈ 0.286) contacts per day (2 times for 7 days). The other parameters were defined by some experiments conducted in this study. To compare the HLD model to the actual dynamics of TBC incidences previously reported, disease incidence data of three different commercial greenhouses (approximately 1000 tomato plants per greenhouse), which had a severe outbreak in 2005 in Okayama, Japan [2,10], were used.

Moreover, the basic reproduction number (*R*_0_) and the effective reproduction number (*R_t_*) of TBC were calculated. The general forms of the equations of *R*_0_ and *R_t_* were given by the following:*R*_0_ = *βd*,(6)
*R_t_* = *R*_0_ (*H*/*N*),(7)

In epidemiology, the *R*_0_ of an infection is the expected number of cases directly generated by one case in a population where all individuals are susceptible to infection, and *R_t_* is the number of cases generated in the current state of a population [20,21]. In commonly used infection models, the infection will be able to spread in a population when *R*_0_ or *R_t_* >1, but not if *R*_0_ or *R_t_* <1. Generally, the larger the value of *R*_0_, the harder it is to control the epidemic. Based on the parameters of the HLD model, *R*_0_ and *R_t_* were estimated.

## 5. Conclusions

This is the first study in which an HLD model of TBC was developed using some biological parameters obtained from precise inoculation experiments. Moreover, the effect of ethanol disinfection of *Cmm*-infected scissors was revealed. Using an epidemiological model like HLD could help us not only understand the relationship between increasing numbers of diseased plants and terms of disease incidence, but also simulate control during cultivation. The HLD can help farmers decide on a plan to control TBC based on their own cultivation timeline. In future studies, we will verify the non-infectious period of latently infected tomato plants and add new parameters to the HLD model to continue to improve its precision. 

## Figures and Tables

**Figure 1 plants-11-02253-f001:**
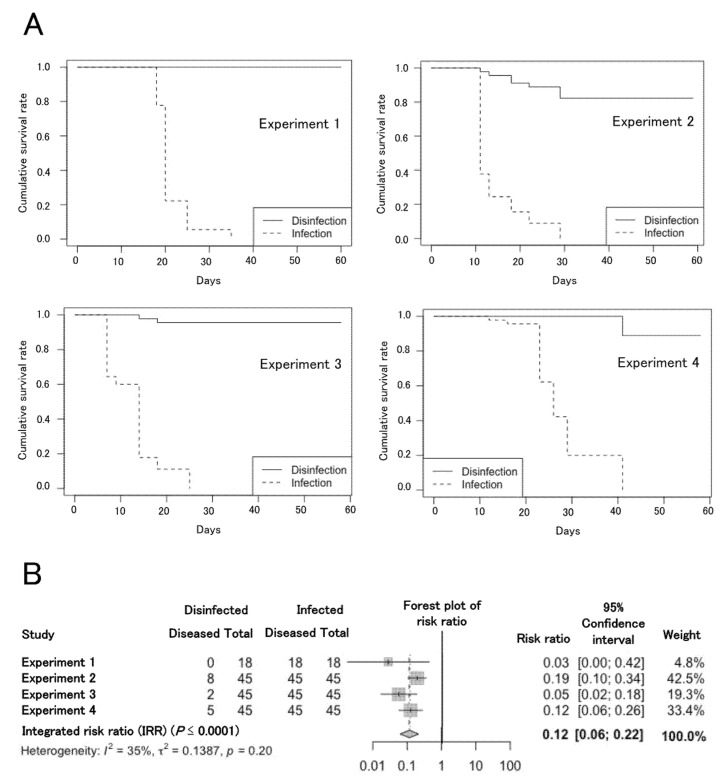
Assessment of effect of disinfected scissors on disease development. (**A**) Kaplan-Meier analysis of plant survival after infection by cutting with disinfected or infected scissors. (**B**) Integrated evaluations based on meta-analyses of effects of cutting with disinfected scissors. The center of each lozenge marks the value of the integrated risk ratio. Each gray square marks the value of the risk ratio (RR), and the size of the square indicates the weight of each experiment. The spread (horizontal line) indicates the 95% confidence interval.

**Figure 2 plants-11-02253-f002:**
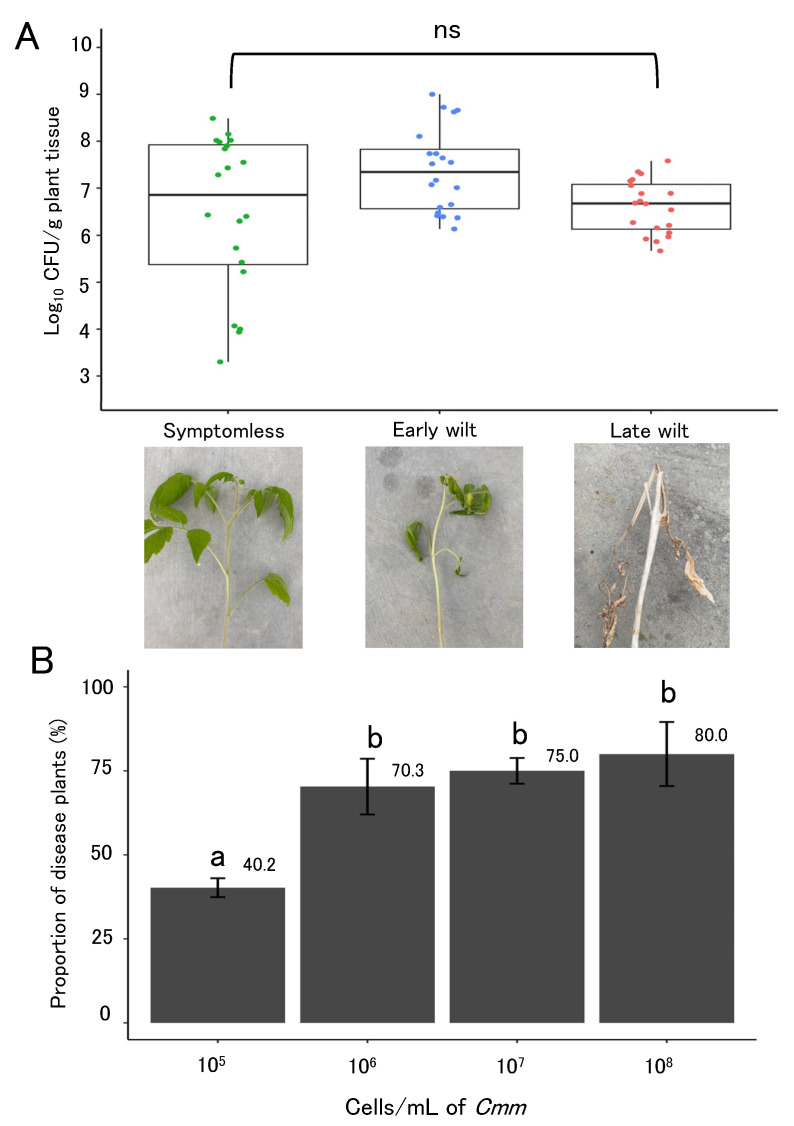
Relationship between populations of *Cmm* and probability of tomato bacterial canker (TBC) infection. (**A**) Populations of *Cmm* in tomato bacterial canker-infected plants, grouped according to symptoms (symptomless, early wilt, and late wilt). The center bar of the boxplot is the median, the lower and upper horizontal bars are the 25th and 75th percentile, and whiskers show the 95% range. The values of log_10_ CFU/g plant tissues with three different symptoms were not significantly different (ns, *p* > 0.05, ANOVA). (**B**) Proportion of diseased plants inoculated with different concentrations of *Cmm*. Error bars show standard error of mean (SEM). Different letters indicate a significant difference from the other bars in different concentrations of *Cmm* (*p* ≤ 0.05, Tukey’s HSD test).

**Figure 3 plants-11-02253-f003:**
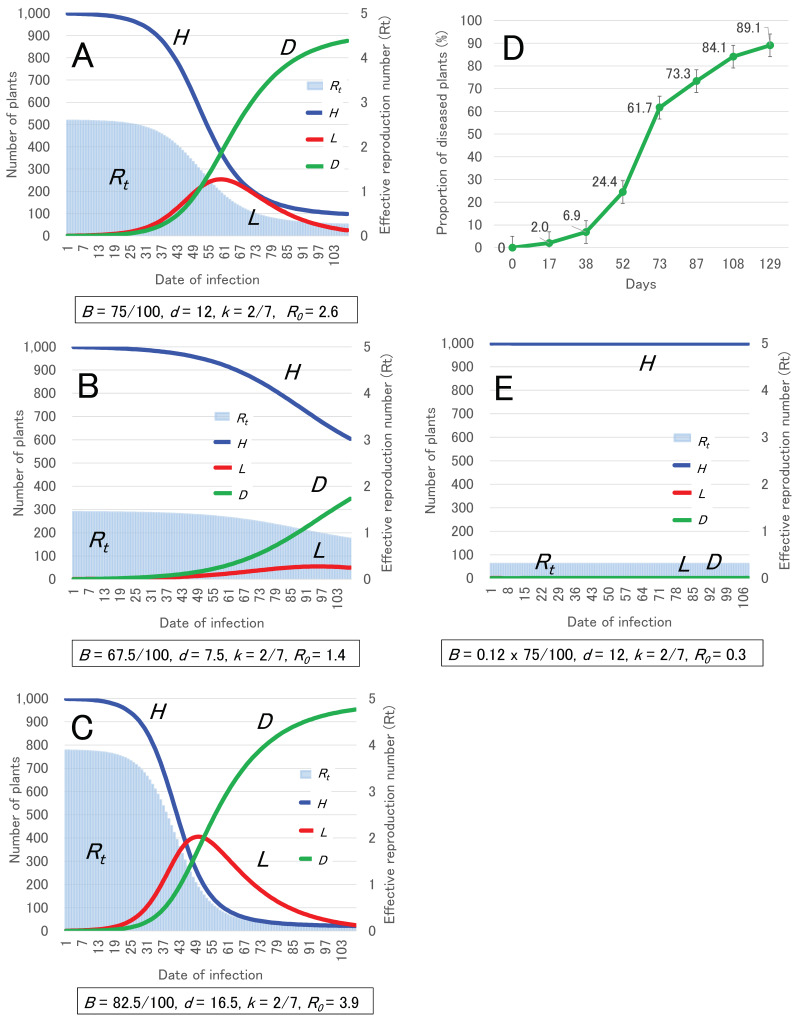
Development of HLD model of TBC and estimation of TBC transmission. (**A**) HLD model A and dynamics of the effective reproduction number (*R_t_*) of TBC. (**B**) model B and dynamics of *R_t_*. (**C**) model C and dynamics of *R_t_*. (**D**) dynamics of TBC incidence actually observed in commercial greenhouses in Okayama, Japan. Data was shown as the mean proportion of TBC incidence (*n* = 3), and error bars show standard error of the mean (SEM). (**E**) estimation of *H*, *L*, *D*, and *R_t_* of TBC in the case of disinfection of scissors based on model A.

**Table 1 plants-11-02253-t001:** Effects of ethanol disinfection on disease development and plant survival.

Experiment	Inoculation	No. of Tomato Seedlings	Proportion of Diseased Plants (%) ^a^	Period between Inoculation and Symptom Development (Days)	Mean of Survival (Days) ^b^	*p* Value ^c^
Experiment 1	Disinfection of infected scissors	18	0	≥60	60.0	9.0 × 10^−10^
	Non-disinfection of infected scissors	18	100	18	21.2	
Experiment 2	Disinfection of infected scissors	45	11.1	11	54.3	2.0 × 10^−16^
	Non-disinfection of infected scissors	45	91.1	11	14.2	
Experiment 3	Disinfection of infected scissors	45	4.4	14	56.1	2.0 × 10^−16^
	Non-disinfection of infected scissors	45	100	7	12.8	
Experiment 4	Disinfection of infected scissors	45	11.1	41	56.1	2.0 × 10^−16^
	Non-disinfection of infected scissors	45	100	12	28.1	
Average	Disinfection of infected scissors	-	6.7	31.5	56.6	5.5 × 10^−5^
	Non-disinfection of infected scissors	-	97.8	12.0	19.1	
	Difference (days)			19.5	37.5	

^a^ Each seedling represented one biological replicate. ^b^ According to Kaplan–Meier survival analysis. Shorter survival indicates quicker disease development. ^c^ According to the log-rank test. The *p* value was calculated for comparison of different inoculation methods. For the average data of mean of survival, the *p* value was calculated by Student’s *t*-test.

**Table 2 plants-11-02253-t002:** Parameters used in HLD model of tomato bacterial canker transmission.

Parameter	Description	Variable	Reference	Assumed Range (95% CI)
*B*	Infection rate per contact	75/100	Figure 2B	67.5/100–82.5/100
*k*	Number of contacts per day	2/7	Assumed	Not applicable
*d*	Duration of infection (days)	12	Table 1	7.5–16.5

## Data Availability

The data presented in this study are available on request from the corresponding author.

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
