# Peer review of "An HLD Model for Tomato Bacterial Canker Focusing on Epidemics of the Pathogen Due to Cutting by Infected Scissors"

_plants, 2022, doi:10.3390/plants11172253_

Round 1
Reviewer 1 Report
Dear Authors, I reviewed the manuscript (plants-1872141) entitled A HLD model for tomato bacterial canker focusing on epidemics of the pathogen due to cutting by infected scissors. This manuscript presents relevant information about findings in tomato bacterial canker. However, some sections of the presented data can be improved. For this reason, I consider that this manuscript needs minor changes to be considered for its publication in this journal.
Additional comments.
Highlight the advantages of using HLD models to evaluate tomato infections.
Check paragraph extension in this manuscript.
Try to detail how the tomato infection was performed.
Include an experimental design containing statistical factors and response variables in the statistical analyses applied to the findings of this research.
Try to compare the obtained findings with similar assays where HDL models were used to evaluate fruit infections.
Include future trends to keep working with the obtained data.
Try to conclude with a general statement of the most relevant part of this study.
Author Response
Additional comments.
- Highlight the advantages of using HLD models to evaluate tomato infections.
Response by Authorsï¼›
Thank you for your suggestion. In conclusions section, we have already described the advantages of using HLD as below,
“Using an epidemiological model like HLD could help us not only understand the relationship between increasing numbers of diseased plants and terms of disease incidence, but also simulate control during cultivation”, which is a compact sentence we tried to avoid becoming redundant. To show the advantage for simulation of positive effect of control, plus, we have added the new sentence in conclusions section as below,
“HLD can help farmers decide the plan of controlling TBC considering on own cultivation term.”
- Check paragraph extension in this manuscript.
Response by Authorsï¼›
Thank you for your suggestion. We have checked paragraph extensions in Introduction section, the paragraph “TBC causes frequent outbreaks…” has been connected to previous paragraph. Moreover, in Materials and methods section, 4.4. Development of HLD model, the paragraph “In this study, some parameters were defined by…” has been connected to previous paragraph.
- Try to detail how the tomato infection was performed.
Response by Authorsï¼›
Thank you for your suggestion. In lines 42 to 45, we have added the new sentences as below; “The source of bacterial canker infection in greenhouses in Japan, consisted of residual plants in the soil as the primary inoculum and disbudding and defoliation with con-taminated equipment and hands as the secondary inoculum [2]. Infected plants increase rapidly due to infection transferred next healthy plants by disbudding and defoliation [2,8].”
- Include an experimental design containing statistical factors and response variables in the statistical analyses applied to the findings of this research.
Response by Authorsï¼›
Thank you for your suggestion. In HLD model based on SIR model, parameters are not estimated from huge number of observation and/or automatic measured data by some regression techniques, maximum likelihood, or Bayesian estimation but obtained from some experiments and previous reported data (described in lines 164-168). Thus, response variables did not have statistical range like 95% CI and were not performed by any statistical analysis to calculate p-values.
- Try to compare the obtained findings with similar assays where HDL models were used to evaluate fruit infections.
Response by Authorsï¼›
Thank you for your suggestion. But I am not sure the mean of “fruit infections”. It might be a canker on fruits like “bird-eye” symptoms? In this study, we defined TBC occurrence in greenhouses avoiding rain and/or storm. We have previously reported that cutting plants contributed to TBC outbreaks in greenhouse. In general, fruit infections do not occur in greenhouses. To evaluate fruit infections, some new parameters should be needed but it will take more time to obtain them by some experiments. We would like to evaluate fruit infection and compare two different models in the next article.
- Include future trends to keep working with the obtained data.
Response by Authorsï¼›
Thank you for your suggestion. We have added the new sentence in Conclusions section as below,
“In future studies, we will verify the non-infectious period of latently infected tomato plants and add new parameters to HLD model to keep updating the model that can stimulate more precisely.”
- Try to conclude with a general statement of the most relevant part of this study.
Response by Authorsï¼›
Thank you for your suggestion. In Conclusions section, the sentence have been moved to in lines 211 to 213 in Discussion section as below,
“Based on these results, it is recommended that growers sterilize their scissors and gloves by dipping them in 70% ethanol to prevent outbreaks of TBC in commercial greenhouses.”
Reviewer 2 Report
Manuscript title: A HLD model for tomato bacterial canker focusing on epidemics of the pathogen due to cutting by infected scissors
Manuscript ID: plants-1872141
Journal: Plants
The main aims of this manuscript were to develop the “HLD” epidemic model based on SIR theory of tomato bacterial canker and to analyze the relationship between disease incidence and infection probability by cutting plants with infected scissors. Also, the efficacy of disinfection of scissors contaminated with y Clavibacter michiganensis subsp. michiganensis to prevent infection of Clavibacter michiganensis subsp. michiganensis by cutting was assessed. The idea is sound and the manuscript is well written. However, several points need to be considered before accepting the current version;
Introduction section is not updated and lacks of the current status of the manuscript idea;
What are the main challenges that the farmers face?
What are the benefits of such model?
Statistical analysis must be performed and revised particularly Table 1;
The quality of the whole figures is poor;
Author Response
- Introduction section is not updated and lacks of the current status of the manuscript idea;
Response by Authorsï¼›
Thank you for your suggestion. We have added more three references in Introduction section.
- Madden, L.V.; Van den Bosch, F. A population-dynamics approach to assess the threat of plant pathogens as biological weapons against annual crops. BioSciense. 2002, 52, 65-74.
- Gilligan, C. A.; Gubbins, S. Analysis and fitting of an SIR model with host response to infection load for a plant disease. Phil. Trans. R. Soc. Lond. B. 1997, 352, 353-364
- Kleczkowski, A.; Hoyle, A.; McMenemy, P. One model to rule them all? Modelling approaches across OneHealth for human, animal and plant epidemics. Phil. Trans. R. Soc. B. 2019, 374, 20180255. http://dx.doi.org/10.1098/rstb.2018.0255.
- What are the main challenges that the farmers face?
Response by Authorsï¼›
Thank you for your suggestion. I think that the main challenge of tomato growers is controlling for TBC. To control TBC efficiently, farmers want to know how long the control methods keep working during the cultivation term. Each cultivation term by individual farmers is different depending on cultivars, greenhouses, open-air fields, fruits shipping term. Some farmers try to cultivate tomatoes for 6 months, but the other might want to do for only 3 months. The HLD model can show how many healthy plants will remain during own cultivation term in case of using a control method because the days in x-axis can be determined by each user. Thus, HLD can help farmers decide the plan of controlling TBC depending on own farming term.
- What are the benefits of such model?
Response by Authorsï¼›
Thank you for your suggestion. The answer is as same as that to the previous question from you. HLD can help farmers decide the plan of controlling TBC considering own farming term. We have added the new sentence in conclusions section as below,
“HLD can help farmers decide the plan of controlling TBC considering on own cultivation term.”
- Statistical analysis must be performed and revised particularly Table 1;
Response by Authorsï¼›
Thank you for your suggestion. In Table 1, log-rank test is a statistical analysis, and we performed that and showed P value in each experiment 1 to 4. For the average of mean of survival (days), t-test as statistical analysis was performed and newly added the P value “5.5 × 10−5” in Table 1. Also, we added the sentence as additional explanation in Table 1 as below,
“For the average data of mean of survival, the P value was calculated by Student’s t-test.”
- The quality of the whole figures is poor.
Response by Authorsï¼›
Thank you for your suggestion. Figs 1 and 2 were made by R package “gglot2”, and we have corrected the character of 'Risk ratio' in Fig 1B. Also, we have corrected the position of values of population of disease plants (%) in Fig 3D.